# Exploratory and confirmatory factor analysis of the new region-generic version of Fremantle Body Awareness—General Questionnaire

**David M. Walton**[1,2]*, **Goris Nazari**[3], **Pavlos Bobos**[1], **Joy C. MacDermid**[1,4]

**1** Faculty of Health Science, School of Physical Therapy, Western University, London, ON, Canada, **2** Department of Psychiatry, Schulich School of Medicine and Dentistry, Western University, London ON, Canada, **3** Queens University, Kingston, ON, Canada, **4** Roth McFarlane Hand and Upper Limb Centre, St. Joseph's Hospital, London, ON, Canada

* dwalton5@uwo.ca

**Data Availability Statement:** Data cannot be shared publicly because of confidentiality requirements. De-identified data are available from

## Abstract

### Background

As the field of pain evaluation grows, newer and more targeted tools are being published for patient-centric evaluation of specific aspects of the pain experience. The Fremantle Back Awareness Questionnaire (FreBAQ) is intended to capture alterations in bodily awareness or perception. To date only region-specific (back, neck, shoulder, knee) versions have been published.

### Objectives

The purpose of our study was to report on the properties of a new region-generic version of the FreBAQ, the FreBAQ-general. Structural validity, internal consistency, and convergent validity against external criteria were evaluated in a sample of Canadian military veterans with chronic pain, with results compared against those published for the region-specific FreBAQ versions.

### Methods

Eligible participants were those that had prior military service, were at least 18 years of age and self-identified as having chronic pain. We used a split-sample approach to Exploratory Factor Analysis (EFA) and Confirmatory Factor Analysis (CFA) on independent random samples. Factor structure, internal consistency, and associations with external criteria were used to compare against prior versions.

### Results

328 respondents (74% of consented) completed at least 7 of the 9 FreBAQ-general questions. EFA and CFA on two independent samples offered support for both 6- and 7-item versions. Comparisons against the external criteria (pain severity, interference,

the senior author (contact via jmacderm@uwo.ca) or the the Western University Office of Research Ethics (contact via ethics@uwo.ca) for researchers who meet the criteria for access to confidential data.

**Funding:** This project was funded by the Government of Canada and the Chronic Pain Centre of Excellence for Canadian Veterans. The funders had no role in study design, data collection and analysis, decision to publish, or preparation of the manuscript.

**Competing interests:** The authors have declared that no competing interests exist.

catastrophizing) indicated no statistical superiority of one over the other, so in the interest of parsimony the 6-item FreBAQ-general was endorsed.

## Conclusions

The Fremantle Body Awareness Questionnaire (FreBAQ-general) showed psychometric properties very much in alignment with those previously reported for the region-specific versions, and sound factorial validity accomplished with fewer items (6 vs. 9). We believe this version can be implemented in practice for those seeking a shorter scale without the need to have multiple region-specific versions on hand, though suggest that those seeking direct comparability with previously published work will still wish to use the original versions.

## Introduction

Trauma-related chronic pain and distress represents tremendous burden in North America. For example, current estimates indicate that the lifetime prevalence of post-traumatic stress disorder (PTSD) amongst Canadian military veterans is 22% [1] compared to 9.2% of the general Canadian population [2]. Of those diagnosed with PTSD, 41% also report comorbid chronic pain [3]. Military veterans represent a population of particular importance for optimizing trauma-related pain management, as they are more likely than the general population to be exposed to both traumatizing experiences and increased biomechanical loads leading to injury and pain.

Management of chronic pain is recognized as a priority area for rehabilitation research by several bodies including the World Health Organization [4,5]. Looking across the current evidence there is little to suggest any one intervention strategy for chronic non-cancer pain is consistently superior to any other [6], and on balance the effects of all chronic pain treatment strategies tend to be small [6]. Adding to the burden are the often-cited side effects of some common pain management strategies, such as opioid use disorder [7], that when considered together indicates a high priority need for better intervention strategies to improve outcomes for Canadians, including veterans, living with trauma-related chronic pain.

Many prior scholars have endorsed a shift away from 'one-size-fits-all' approaches to pain management in favour of more person-specific or precision treatment paradigms [8]. For this to be realized, rigorous evaluation procedures that can facilitate 'phenotyping' or mechanism-based classification are needed. As the field of pain evaluation grows, newer and more targeted tools are being published that permit patient-centric evaluation of specific aspects of the pain experience. One such tool is the Fremantle Back Awareness Questionnaire (FreBAQ, [9], accessible at https://oml.eular.org/sysModules/obxOML/docs/id_412/wand2014_questionnaire.pdf). Embedded within a theoretical framework of sensorimotor disembodiment often described by a subgroup of people with chronic pain (e.g., [10]), prior work has found that the experience of bodily (un)awareness is associated with reports of pain, disability, and catastrophic pain beliefs [11]. Evidence drawn largely from qualitative work also indicates that a sense of kinesthetic 'wholeness' [12] or trialectic body-self-society integration [13] may represent a theoretically sound treatment outcome, lending greater value to tools such as the FreBAQ. This suggests that for at least a subgroup of people with chronic pain the sense of disembodiment or quasi-neglect of the painful body part may be contributing to, or caused by, the experience of pain. Despite generally low-quality evidence, recent work has demonstrated potential value in treatments aimed at sensorimotor awareness/reintegration strategies for

managing chronic pain [14,15], so a sound clinical evaluation tool appears to be a valuable potential addition to a broader pain evaluation battery.

The FreBAQ has been translated into several languages [10] and has since been adapted to versions specific to the neck (FreNAQ, [10]), shoulder (FreShAQ, [16], and knee (FreKAQ, [17]). A systematic scoping review from Viceconti and colleagues [15] found that the FreBAQ and its region-specific derivatives were used in the overwhelming majority of studies exploring bodily self-perception, suggesting researchers and clinicians see value in its use. While all versions have demonstrated sound psychometrics, the region-specific nature of each requires clinicians to keep a version of each tool on hand and their interpretation metrics implemented based on the presenting patient's complaint. Arguably a desirable option would be a single region-generic tool that can be implemented regardless of affected body region, as in contexts such as injury from military action that can affect many regions of the body. Electronic data collection platforms make this more feasible as text-piping logic allows the patient to first enter the most affected body region that is then auto-populated through the relevant items. The purpose of this study is to report on the properties of a new region-generic version of the FreBAQ called the FreBAQ-general. Structural validity, internal consistency, and convergent validity against external criteria was evaluated in a sample of Canadian military veterans with chronic pain, and results were compared against those published for the other region-specific versions.

## Materials and methods

### Procedure

Data for this analysis were collected through an online survey of Canadian military veterans that ran from 2021-December to 2022-April. The survey was distributed through an electronic platform (Qualtrics) representing a total target population of 617,800 veterans [18]. The survey was conducted according to standards of best practice for survey research per the Checklist for Reporting Results of Internet e-Surveys (CHERRIES) framework [19]. This included partnership between researchers and representatives of the target population, pilot and field testing of the prototype survey, revision and refinement to optimize rigour of the data collected while minimizing burden on respondents. Access to the survey was through clicking a link that presented the Letter of Information and conducted the eligibility screening through routing logic before presenting the main survey. Permission to modify the scale for the Canadian military veteran population was obtained from the author of the original FreBAQ (personal communication). All methods were approved by the Health Sciences Research Ethics Board of Western University (London Ontario, Canada) prior to enrolling participants. The survey was an amalgam of tools, tests, and questions, some established and standardized, some (like the FreBAQ-general) adapted from established tools, with some study specific questions related largely to demographics and experiences while in the military (e.g., years of service, rank at release, types of trauma exposures). Eligible, consenting participants completed the questionnaire items independently in a fixed order, and at the end were invited to also participate in a follow-up qualitative interview that will be reported in a separate manuscript.

### Participants

Eligible participants were those that had prior military service but were no longer actively enrolled (e.g., retired, discharged, or left for other reasons), were at least 18 years of age, could understand conversational (grade 6) English or French, and self-identified as having chronic pain (definition left intentionally broad).

## Outcome measures

**Fremantle Body Awareness Questionnaire (FreBAQ-general).** The original FreBAQ is a 9-item self-report tool that includes items such as 'My back feels like it is not part of the rest of my body' and 'My back feels lopsided or asymmetrical' with each item rated on a 5-point frequency-based scale from 'never' to 'always'. Prior work on the FreBAQ and its other region-based analogs (FreShAQ, FreKAQ, FreNAQ) have consistently found a single factor structure with Cronbach's alpha at or near the 0.80 level [20,21]. Accordingly, the total score is most typically summed and reported out of 36 where a higher number represents a greater sense of bodily disconnection / unawareness in the affected region. For the FreBAQ-general, our team created a system in the online survey in which respondents were first asked 'On an average day, what is your <u>most painful</u> body part?' with an open text response. Responses were then piped into the relevant part of the FreBAQ items, so the questions given as examples above were adapted to read 'My [body part] feels like it is not part of the rest of my body' and 'My [body part] feels very lopsided or out of proportion to that on the opposite side' where [body part] was replaced verbatim with the respondents response to the prior question. Researchers on this study (DW, JM, PB) are experienced psychometricians with a long track record of patient-reported scale development, and the prototype version was included in the pilot field testing for feedback and revision before deploying to the broader population.

*Brief Pain Inventory (BPI).* The BPI [22] is one of the most widely-used pain-specific patient reported outcomes (PROMs) and has been translated into several languages (incl. English and French) [23]. For this study only the quantifiable Pain Severity (mean of 4 items rated 0–10, Cronbach's $\alpha$ in current sample = 0.87) and Pain Interference (7 items rated 0–10, Cronbach's $\alpha$ = 0.93) sections were used. Each section has consistently demonstrated sound internal consistency with the Interference subscale being reportable as either a single summative score or at least two subscales (Physical and Affective Interference, [22]). In both subscales, a higher number indicates more severe pain or greater pain-related functional interference. While originally designed for use in people with cancer pain, it has since been used extensively in non-cancer chronic pain research [24].

*The Brief Pain Catastrophizing Scale 4 item version (BriefPCS-4).* The BriefPCS-4 is an abbreviated version of the original 13-item PCS [25,26] that has previously demonstrated sound structural, internal, and concurrent validity indicators for use in research involving people with chronic pain [27]. It is intended to capture exaggerated negative orientation towards pain [25] using a 5-level response structure from 'not at all' to 'all the time'. In prior work, this 4-item version correlated with the full version at r > 0.95 [27] indicating that scores on the brief version are reflective of what they would have been on the longer version, while minimizing burden in what was a lengthy online survey. Cronbach's $\alpha$ in the current sample is calculated at 0.89.

*Other Demographic Data.* Other data included those necessary for describing the participants, including sex-at-birth, current age, educational attainment, rank at release, length of service, primary service element (army, navy, air force, other), current employment status, and primary mechanism(s) of injury. For the majority of respondents study participation was complete after the single administration of the survey.

## Statistical analysis

### Descriptives

Demographics and clinical pain indicators (BPI Pain Severity and Pain Interference, BriefPCS-4) were summarized descriptively. Response distributions on the FreBAQ-general were evaluated for floor or ceiling effects (defined as $\geq$50% of the sample selecting the lowest

or highest option), normality, and missing values. Where 2 or fewer (out of 9) values were missing those were replaced with the mean of the answered items, while those with 3 or more missing values were removed from analysis.

**Exploratory factor analysis.**   The sample was split into one training and one validation subset. For the training subset we followed the advice of Costello and Osborne (2005) [28] of a 20:1 respondent: item ratio, extracting n = 180 responses at random. We used direct oblimin rotation on principal axis factor-exploratory factor analysis (PAF-EFA) as we had no expectation of normal distribution of responses to each variable. The number of factors to be extracted was determined using Velicer's minimum average partial (MAP) [29] test that iteratively partials out common variance between components until only unique variance is left, the number of steps to do so being the number of factors to be extracted [30]. Individual item fit was explored through communalities at extraction, with those <0.40 considered for further exploration and possible removal [31]. A sound model was considered one in which at least 50% of overall scale variance could be explained by the items, no factor loading <0.32 and no item loaded across more than one factor at > 0.32 [32].

**Confirmatory factor analysis.**   The remaining n = 148 responses were used as the validation set through Confirmatory Factor Analysis (CFA) in which the structure from the EFA was recreated and tested on an independent sample. Standard goodness-of-fit indicators and fit residuals were used to interpret these results: comparative fit index (CFI) > 0.90 [33], Tucker-Lewis Index (TLI) > 0.90, root mean square error of approximation (RMSEA) < 0.08 [34], and standardized root mean square residual (RSMR) < 0.05 [34]. Where the indicators suggested sub-optimal fit, residuals and modification indices were explored to identify potentially problematic items or location dependence between two or more items. Where a potentially problematic item was identified, it was removed and the model retested, followed by re-evaluation of the remaining items in the EFA with the first random set.

**External validation.**   Equivalence with previously published FreBAQ scales was evaluated through comparison of correlation coefficients between FreBAQ-general and the three external criteria: the BPI Severity and Interference subscales, and BriefPCS-4 using bootstrapped 95% confidence intervals for Pearson's r. A fourth criterion was Cronbach's alpha of the FreBAQ-general. The comparators were those reported by Wand and colleagues, who found a correlation between FreBAQ and PCS scores of r = 0.36, FreBAQ and Numeric Pain Rating Scale (NPRS) scores of r = 0.27, FreBAQ and low back disability scores of r = 0.32, and Cronbach's alpha = 0.80 [9]. While the populations and tools for evaluating pain severity and interference were different between that study and ours, we hypothesized that all indicators in our data should *at minimum* be significant, in the same direction (i.e., positive), and of similar magnitude to those reported by Wand and colleagues. Inclusion of one point estimate within the 95%CI of the other would strengthen confidence in equivalence but could not be considered essential considering the differences between the two studies.

All available data were used for these analyses using Mplus version 6.14 (Muthen & Muthen).

## Results

### Descriptives

Electronic online consent to participate was provided by 445 Canadian veterans, of which 328 (74%) completed at least 7 of the 9 FreBAQ-body questions on the survey. Respondents were 66% male, mean age of 54 years (range 26 to 84), and most (58%) reported 20 years or more of military service (Table 1). The sample reported overall moderate BPI Pain Severity (mean = 5.8, SD 1.6) and moderate-to-high BPI Pain Interference (mean = 6.7, SD 2.3; Table 1).

**Table 1. Characteristics of participants in the entire sample.**

| N = 328 | Frequencies (%) |
|---|---|
| **Sex** (% female) | 117 (34%) |
| **Age** (mean, range) | 54 (26 to 84) |
| **Education** | |
| Did not finish high school | 14 (4%) |
| High school diploma | 30 (10%) |
| Did not finish college | 50 (15%) |
| College, trade, or technical school diploma | 111 (35%) |
| University undergraduate degree | 68 (21%) |
| University graduate degree | 46 (14%) |
| Other | 9 (2%) |
| **Rank at Release** | |
| Commissioned Officer | 60 (23%) |
| Non-Commissioned Officer | 78 (32%) |
| Non-Commissioned Member | 181 (43%) |
| Other | 9 (2%) |
| **Total length of Service** | |
| <5 years | 23 (6%) |
| 5–9 years | 34 (9%) |
| 10–19 years | 111 (26%) |
| 20 years or more | 160 (58%) |
| **Primary Service Element** | |
| Army | 200 (61%) |
| Navy | 52 (16%) |
| Air Force | 73 (22%) |
| Other | 3 (1%) |
| **Current Employment Status** | |
| Still employed by CAF | 55 (17%) |
| Other, full-time | 52 (16%) |
| Other, part-time | 25 (7%) |
| Unable to work, disability | 145 (44%) |
| Unable to work, other | 7 (2%) |
| Retired or choosing not to work | 49 (14%) |
| **Mechanism of Service Injury** | |
| Explosion | 16 (5%) |
| Gunfire | 1 (<1%) |
| Hand-to-hand Combat | 2 (1%) |
| Training | 75 (23%) |
| Fall | 51 (15%) |
| Overuse/Strain | 89 (27%) |
| Unknown | 14 (4%) |
| Other | 80 (24%) |

CAF = Canadian Armed Forces.

Evaluation of score distribution across the 9 FreBAQ-body items revealed that item 8 'My [body part] feels smaller than it should be' showed strong evidence of a floor effect (64% selected 'never') while no items showed a ceiling effect. Items 1 'My [body part] feels as though it is not part of the rest of my body', 6 'I can't perceive the outline, or borders, of my [body part]', and 7 'My [body part] feels larger than it should be' showed evidence of a bimodal distribution spread between 'never' and 'occasionally'/'often' responses. Table 2 shows the pattern of responses. At this point no items were removed.

## Exploratory and confirmatory factor analysis

After extracting a random sample of n = 180 (mean age = 52.9 years, 34.8% female), the MAP test indicated a single factor should be extracted. The initial factor structure including all 9

**Table 2. Response frequencies of the 9 FreBAQ-general items.**

| | Never | Rarely | Occasionally | Often | Always | Median | Mean |
|---|---|---|---|---|---|---|---|
| 1. My [body part] feels as though it is not part of the rest of my body | 84 (26%) | 39 (12%) | 52 (16%) | 92 (28%) | 61 (19%) | 2 | 2.0 |
| 2. I tend to focus all my attention on my [body part] to make it move the way I want to | 20 (6%) | 20 (6%) | 20 (6%) | 144 (44%) | 82 (25%) | 3 | 2.8 |
| 3. I feel as if my [body part] sometimes moves involuntarily, without my control | 82 (25%) | 72 (22%) | 95 (29%) | 52 (16%) | 26 (8%) | 2 | 1.6 |
| 4. When performing everyday tasks, I don't know how my [body part] is moving | 75 (23%) | 75 (23%) | 62 (19%) | 92 (28%) | 23 (7%) | 2 | 1.7 |
| 5. When performing everyday tasks, I am not sure what position my [body part] is in | 88 (27%) | 84 (26%) | 66 (20%) | 56 (17%) | 33 (10%) | 1 | 1.6 |
| 6. I can't perceive the outline, or borders, of my [body part] | 134 (41%) | 56 (17%) | 62 (19%) | 46 (14%) | 33 (10%) | 1 | 1.4 |
| 7. My [body part] feels larger than it should be | 111 (34%) | 46 (14%) | 56 (17%) | 62 (19%) | 49 (15%) | 2 | 1.7 |
| 8. *My [body part] feels smaller than it should be* | *210 (64%)* | *62 (19%)* | *30 (9%)* | *13 (4%)* | *13 (4%)* | *0* | *0.7* |
| 9. My [body part] feels very lopsided, or out of proportion to that on the opposite side | 95 (29%) | 39 (12%) | 69 (21%) | 59 (18%) | 69 (21%) | 2 | 1.9 |

*Italics*: Item with strong evidence of floor effect.

items revealed acceptable fit though did not reach the 50% of variance explained threshold (KMO = 0.84, Sphericity $\chi^2$ = 540.2 p < 0.001, 47.4% of variance explained). Two items showed low communalities: item 2 "I tend to focus all my attention on my [body part] to make it move the way I want to" and item 8 "my [body part] feels smaller than it should be" had communalities of 0.22 and 0.23, respectively (Table 3). Considering the goal of parsimony, we evaluated the properties of the tool with those two items removed, thereby bringing a 7-item version forward. This shortened version resulted in a small but potentially important improvement in variance explained while not adversely affecting other indicators (KMO = 0.86, Sphericity $\chi^2$ = 456.7 p < 0.001, 54.1% of variance explained).

This 7-item version was then tested through CFA using the responses of the remaining n = 148 (mean age = 55.3 years, 33.5% female). Fig 1A shows the factor loadings of the 7 items

**Table 3. Communalities at extraction of the original 9 items of the FreBAQ-general, and principal axis factoring-based loadings of the retained 7 items.** Items are displayed in order of decreasing factor loading.

| | Communality at Extraction | Factor Loading |
|---|---|---|
| When performing everyday tasks, I am not sure exactly what position my [body part] is in | 0.70 | 0.85 |
| I can't perceive the outline, or borders, of my [body part] | 0.59 | 0.78 |
| When performing everyday tasks, I don't know how my [body part] is moving | 0.58 | 0.78 |
| I feel as if my [body part] moves involuntarily, without my control | 0.39 | 0.60 |
| My [body part] feels larger than it should be | 0.31 | 0.59 |
| My [body part] feels as though it is not part of the rest of my body | 0.37 | 0.58 |
| My [body part] feels very lopsided, or out of proportion to that on the opposite side | 0.36 | 0.57 |
| My [body part] feels smaller than it should be | 0.23 | |
| I tend to focus all my attention on my [body part] to make it move the way I want to | 0.22 | |

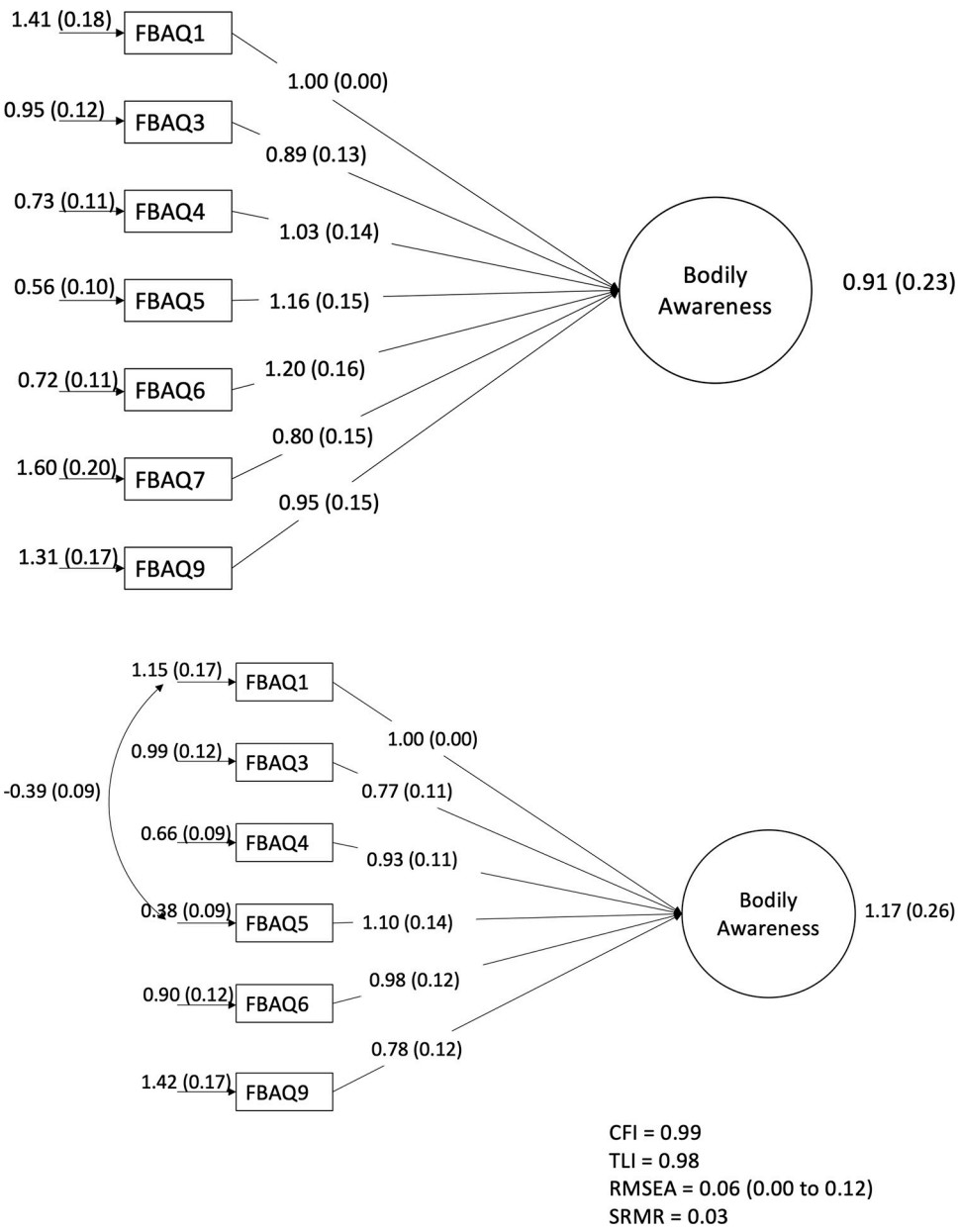

**Fig 1.** a: Base model for confirmatory factor analysis (CFA) including all of the original FreBAQ items. Values are path coefficients (standard error). b: Final CFA model after removal of poorly-fitting items. Values are path coefficients (standard error).

on a single latent construct of 'bodily awareness'. Only two of the four fit indicators of this model met the a priori thresholds for acceptability (CFI = 0.92, TLI = 0.88, RMSEA = 0.13, SRMR = 0.05). Item 7 "My [body part] feels larger than it should be" showed the lowest path coefficient and largest residual with no clear option for improved fit through modification indices. Having already removed its companion ('feels smaller') item, that item was removed and the remaining 6 items re-tested on a single latent construct. Doing so led to acceptable fit on all but the RMSEA indicator (CFI = 0.94, TLI = 0.91, RMSEA = 0.13, SRMR = 0.04), reaching a level of acceptable fit when the residuals of items 1 and 5 were correlated (CFI = 0.99,

**Table 4. Pearson's correlation coefficients with 95% confidence intervals (columns 2–4), and Cronbach's alpha (column 5) for the 6- and 7-item versions of the Fre-BAQ-general against the external criteria.**

|  | BPI Severity | BPI Interference | BriefPCS | Alpha | Omega |
|---|---|---|---|---|---|
| **FreBAQ7** | 0.43 (0.34, 0.51)** | 0.44 (0.35, 0.52)** | 0.36 (0.26, 0.45)** | 0.86 | 0.88 |
| **FreBAQ6** | 0.42 (0.33, 0.51)** | 0.44 (0.36, 0.53)** | 0.37 (0.28, 0.46)** | 0.85 | 0.87 |

BPI = Brief Pain Inventory; BriefPCS = 4-item version of the Pain Catastrophizing Scale. **: Correlations were significant at the p<0.01 level.

TLI = 0.98, RMSEA = 0.06, SRMR = 0.03; Fig 1B). There were no other modifications available to improve model fit.

After EFA and CFA on two independent samples, we were left with some ambiguity on the best version: both a 7-item and a 6-item version demonstrated some statistical support. From here we focused on the external criteria using the full sample to select the best version that balanced psychometrics with parsimony. Table 4 presents the correlation coefficients with the three external outcomes (BPI Severity, BPI Interference, BriefPCS-4) including both Cronbach's alpha and omega values to account for deviation in tau-equivalence. All 4 such indicators were nearly identical between the two versions. Foregrounding the desire for brevity, the 6-item FreBAQ-general was finally endorsed.

## Discussion

We have described the evaluation of a region-generic version of the Fremantle Back Awareness Questionnaire through a process of methodologic triangulation including exploratory and confirmatory factor analyses on two independent sub-samples of Canadian military veterans with chronic pain, including comparison with previously published region-specific versions through four external criteria. This version, termed the 'Fremantle Body Awareness Questionnaire' (FreBAQ-general) shows psychometric properties very much in alignment with those previously reported for the region-specific versions, and sound factorial validity accomplished with fewer items (6 vs. 9). We believe this version can be implemented in practice for those using electronic data capture seeking a shorter scale without the need to have multiple region-specific versions on hand, though suggest that those seeking direct comparability with previously published work will still wish to use the original versions.

We note both similarities and differences between the FreBAQ-general and published versions of the FreBAQ, including those for the low back [9,20], knee [17], neck [10,21], and shoulder [16]. Similarities are the estimates of internal consistency via Cronbach's alpha, and correlations between pain severity, pain-related interference, and pain catastrophizing. At α = 0.85, internal consistency of the 6-item FreBAQ-general was similar or better than those previously reported for the FreBAQ (α = 0.80 [20]), FreSHAQ (α = 0.71 [16]), FreKAQ (α = 0.88), and FreNAQ-J (α = 0.81 [10]). The most consistent external criterion across studies has been the PCS, with the correlation of the FreBAQ-general (r = 0.37) also similar to those of previously published region-specific versions (r/rho ranging from 0.36 FreBAQ [9,20] to 0.70 FreKAQ [17]).

Different is that we had to remove 3 items to achieve satisfactory factor structure, with two remaining items showing potential location dependence. We note that items 7 ('feels larger') and 8 ('feels smaller'), removed from the 6-item FreBAQ-general, have also demonstrated problems with model fit in prior work. [10,16,17,20] The 4 prior versions have also published evidence of a strong floor effect for item 8, and development studies for the FreSHAQ [16] and FreNAQ-J [10] have shown evidence of a floor effect for item 7. It is interesting that the Rasch analyses of these scales have not revealed evidence of location dependence, as intuitively it

would seem that the two share a non-random interpretive bias–that is, a participant who endorses their back feeling larger than it should seems unlikely to *also* endorse that same region feeling smaller than it should, and vice versa. Perhaps this is because the phenomenon of sensorimotor bodily awareness is complex and/or temporally-specific, in that at times the affected part feels larger and at other times smaller than it should, or perhaps this is due to statistical artifact in the prior Rasch analyses due to the strong floor effect or poor targeting of scale items to persons (e.g., Fig 3 in FreSHAQ [16]), Fig 2 in FreNAQ-J [10]). Regardless, that item 9 'my [body part] feels lopsided or asymmetrical compared to that of the opposite side' has been retained with sound model fit suggests that perhaps the construct of bodily awareness is not so much dependent on whether the body region feels larger or smaller per se, but simply that it is perceived *differently* that is the key observable phenomenon.

We also note the three items removed appear to share a conceptual similarity in that all are related to a sort of thematic hyper-awareness of the affected body part (feeling larger, smaller, or focusing all attention). In contrast, the remaining 6 items are conceptually related more to disembodied detachment or sensorimotor dys-integration of the affected body part. One interpretation is that the 3 items removed could represent a second latent construct, and the concept of bodily awarenesss that the FreBAQ purports to measure includes both hyper- and hypo-awareness subscales where the three items we removed are part of an incomplete hyper-awareness subscale. We attempted to create a two-factor CFA model but it led to a more complex model structure, difficulty in achieving convergence, and no notable improvement in model fit or performance over the single factor described herein (not shown). Comparing means and medians of Table 2 to those of the four prior versions (e.g., Wand and colleagues Table 2, pg. 1006), we note our sample overall scored consistently higher (worse) across almost all items. It is therefore also possible that the concept of disembodiment or distorted self-perception functions differently in more severely affected Canadian military veterans compared to those of, for example, ambulatory Australian civilians. We note that the prior Rasch analyses have all endorsed a unidimensional structure, which seems at odds with the notion of a secondary factor, so perhaps the differential item functioning hypothesis is more correct. Regardless, repetition of this structure in other samples will further identify potential differences in model functioning across populations.

While the properties described herein appear to support the generic FreBAQ-general for implementation, we note several properties remain outstanding. Per COSMIN criteria for example, we did not attempt to establish content or face validity of the items, relying instead on the considerable prior work already done on the FreBAQ and its subsequent versions. Further, we have not attempted to explore cross-cultural validity, and while the FreBAQ and its derivatives have already undergone considerable adaptation work [10,16,17,20], we cannot endorse the FreBAQ-general as confidently usable outside of the English-speaking context. Most notably however is that without a gold standard for 'bodily awareness', the FreBAQ-general, like many pain-related PROMs, suffers from an inability to endorse criterion-related validity against an external objective standard. Arguably the closest efforts in this regard are those of Goossens and colleagues [11] who found a significant correlation between FreBAQ scores and the distance by which representation of the lower back in the secondary somatosensory cortex was shifted laterally in both those with and without non-specific low back pain observed through functional magnetic resonance imaging (fMRI). While we continue to wait for a criterion standard 'marker' of bodily awareness, this initial evidence at least supports the contention that perception of bodily awareness as reported on the FreBAQ is cross-sectionally associated with cortical activity in brain regions thought to be responsible for somatotopic representation of the body.

Finally we acknowledge the need to correlate residuals between items 1 "My [body part] feel as though it is not part of the rest of my body" and 5 "When performing everyday tasks, I am not sure what position my [body part] is in" to achieve acceptable fit in the CFA analysis. This suggests that the residuals are not random but are linked, or dependent, in some way. This is not a trivial concern as it violates the assumption of random observations used in Classical Test Theory-based analyses and could have led to an over-saturated model, though the triangulation procedures herein indicate the model appears to be robust and in practice this is unlikely to affect score interpretation. It does suggest that the latent construct informing both is similar and therefore one could arguably be removed, though having already gone from 9 to 6 items we did not feel compelled to shorten further and risk an adverse effect on internal consistency or responsiveness. The latter is particularly germane especially if the scale is to be used to evaluate change over time, as scales with fewer items also allow fewer levels of gradation for discriminating between change/no change [35]. Without the benefit of repeated measures in this study we would be unable to evaluate the effects on responsiveness of removing items further, so made the intentional and informed decision to retain both though flag this as a potential opportunity for further refinement in future iterations of the scale.

While the moderate-to-strong associations between FreBAQ-general and the external pain indicators (severity, interference, catastrophizing) appear to support clinical utility of the tool, we note that the majority of previously published work on the FreBAQ and its analogs have been on psychometrics and region/cultural translation and validation [21]. Importantly, there have been very few intervention studies in which the FreBAQ versions have been used as a core outcome. In one of the few we can find, Miyachi and colleagues included the FreBAQ in a randomized trial of lumbar motor control training vs general trunk training in a sample of 19 participants with chronic low back pain. In that small study, the motor control group showed a significant pre-post improvement in FreBAQ scores but no between-group differences at either the pre- or post-training period. Whether the FreBAQ or its analogs measures an important, and changeable, phenomenon, and what should be considered important change, are questions that require considerably greater empirical work.

## Limitations

Inability to support criterion-related validity, cross-cultural applicability, or responsiveness aside, perhaps the most relevant limitation is related to how the items were adapted for the FreBAQ-general. This was done through a straightforward process of swapping the body region of prior versions (e.g., back, knee, shoulder, neck) with whatever region the respondent indicated was their most bothersome in a prior question. This means for example that, dependent on the body part, a feeling that it was 'swollen' (item 7) may not be relevant, so we removed allusion to a perception of swelling instead retaining just the perception of largeness, same as the analogous item on the FreSHAQ. This also meant that dependent on how the respondent answered the prior question on most bothersome body part, the items may not have grammatically flowed (e.g., entering 'my knee' in the prior question would have meant item 1 read "My my knee feels as though it is not part of the rest of my body'), and any spelling errors would have been carried into the scale items that may have affected respondents' ability to attend to the meaning of each. The rigorous analytic approach and overall statistical similarity with previously published versions mitigates this concern somewhat, though future studies may wish to monitor and ensure responses to the most bothersome body part question grammatically flow and are spelled correctly. Further, the need for automatic survey logic to pipe text from a prior question into subsequent questions means this version is most easily administered through a digital (e.g., computerized) platform rather than traditional paper-and-pencil, thus

accessibility may be limited to those with access to computerized data collection platforms. Finally, the French translation was conducted by a trained bilingual speaker but was not subject to a fulsome cross-cultural translation study to ensure semantic and measurement invariance. While providing a French option was a requirement from an inclusivity perspective in this study of Canadian veterans, we note that only 9 of the 328 respondents in the final sample, or 2.7%, elected to use the French version of the survey. A sensitivity analysis conducted without those 9 responses provided nearly identical results to what has been reported here, though we note that the French version has not undergone formal cross-cultural adaptation and should accordingly be used with caution.

## Conclusions

Through a process of methodologic triangulation, including exploratory and confirmatory factor analyses and convergent associations with relevant external criteria using a large cohort of Canadian military veterans with chronic pain, we have found that a body-region-generic version of the original back-specific FreBAQ tool for capturing perceptions of bodily awareness functions well for clinical and research use. That our sample appeared to be more severely affected than what might be seen in regular civilian practices may affect generalizability of the findings, though the strong model fit and consistent findings with those published for prior versions supports the use of the FreBAQ-general as a single tool that can be administered to patients regardless of body region affected. We believe that having such a tool will optimize clinicians' ability to implement standardized evaluation of bodily awareness without the need to have multiple region-specific versions of the tool on hand or to be limited to just those body regions that have been published. Cross-cultural translation, content validation, and reliability indices are priorities for future work.

## Acknowledgments

The authors wish to thank Margaret Lomotan, Christina Ziebart, and Katrina Munro for study coordination and support.

## Author Contributions

**Conceptualization:** David M. Walton, Joy C. MacDermid.

**Data curation:** David M. Walton, Joy C. MacDermid.

**Formal analysis:** David M. Walton, Pavlos Bobos.

**Funding acquisition:** David M. Walton, Joy C. MacDermid.

**Investigation:** David M. Walton, Joy C. MacDermid.

**Methodology:** David M. Walton, Joy C. MacDermid.

**Project administration:** David M. Walton, Joy C. MacDermid.

**Resources:** Joy C. MacDermid.

**Supervision:** Joy C. MacDermid.

**Validation:** David M. Walton, Goris Nazari, Pavlos Bobos, Joy C. MacDermid.

**Writing – original draft:** David M. Walton, Goris Nazari, Pavlos Bobos, Joy C. MacDermid.

**Writing – review & editing:** David M. Walton, Goris Nazari, Pavlos Bobos, Joy C. MacDermid.

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
