## [Decision Letter · Decision Letter 0]

5 Oct 2022

PONE-D-22-16116Exploratory and Confirmatory Factor analysis of the new region-generic version of Fremantle Body Awareness - General QuestionnairePLOS ONE

Dear Dr. Walton,

Thank you for submitting your manuscript to PLOS ONE. After careful consideration, we feel that it has merit but does not fully meet PLOS ONE’s publication criteria as it currently stands. Therefore, we invite you to submit a revised version of the manuscript that addresses the points raised during the review process.

We look forward to receiving your revised manuscript.

Kind regards,

Jane Elizabeth Aspell, PhD

Academic Editor

PLOS ONE

Journal Requirements:

Funded by the Government of Canada and the Chronic Pain Centre of Excellence for Canadian Veterans.

Reviewers' comments:

Reviewer's Responses to Questions

**Comments to the Author**

1. Is the manuscript technically sound, and do the data support the conclusions?

Reviewer #1: Yes

Reviewer #2: Yes

2. Has the statistical analysis been performed appropriately and rigorously? 

Reviewer #1: Yes

Reviewer #2: Yes

3. Have the authors made all data underlying the findings in their manuscript fully available?

Reviewer #1: Yes

Reviewer #2: No

4. Is the manuscript presented in an intelligible fashion and written in standard English?

Reviewer #1: Yes

Reviewer #2: Yes

5. Review Comments to the Author

Reviewer #1: Thank you for the opportunity to review this work, which reports on the psychometric evaluation of a novel generic version of the Fremantle Body Awareness Questionnaire in a sample of Canadian military veterans with chronic pain. I believe the work will make a useful contribution to literature on body perceptions in chronic pain. However, I have some recommendations for the authors to consider, which mainly centre on the need for greater detail and clarity in reporting. It remains unclear whether the measures were completed in one language (English) or two (English and French), and I have major concerns regarding sematic equivalence and measurement invariance if the method involved the latter.

Introduction

The systematic scoping review by Viceconti et al., 2020 may be useful for guiding some of the introductory comments on body perception and chronic pain, which is currently fairly limited given the content of the questionnaire. There is a large section of the review which focuses on findings from various versions of the Fremantle questionnaire. The review may also be helpful for explaining the later difficulties encountered by the authors with perception of body size for pain-affected body parts.

Reference:

Viceconti, A., Camerone, E. M., Luzzi, D., Pentassuglia, D., Pardini, M., Ristori, D., ... & Testa, M. (2020). Explicit and implicit own's body and space perception in painful musculoskeletal disorders and rheumatic diseases: A systematic scoping review. Frontiers in Human Neuroscience, 14, 83. https://doi.org/10.3389/fnhum.2020.00083

Sections: Eligibility criteria and procedure; Participants

- There was some confusion in the organisation of these sections. For example, there was no information in the “eligibility criteria and procedure” section regarding inclusion/exclusion criteria, and some procedural information (e.g., survey contents) was included in the “Participants” section. I also found it strange that participant N and characteristics were included in the Results section, and not in this dedicated Participants section.

- I noted that inclusion criteria included comprehension of English or French. However, it is not clearly stated whether the survey was delivered in both languages. This is a crucial issue because there are best-practice procedures which should be adhered to when preparing a novel translation to ensure that semantic equivalence, and the newly translated questionnaire requires additional psychometric validation (e.g., Swami & Barron, 2019). The English and French versions of the questionnaires may not demonstrate measurement invariance, and it is not appropriate to combine or compare scores from two different versions of the questionnaire until measurement invariance has been established (e.g., see Chen, 2008). This is particularly important for the novel FreBAQ-general, but the point also applies to all other questionnaire measures included within the survey package.

- What does it mean “Participants completed the balance of the questionnaire independently”? Additional procedural information is required (i.e., did participants complete the survey at home? Was any guidance or interaction with the research team involved?) Furthermore, were participants offered any form of remuneration? How long did the study take to complete on average? Was any debriefing information provided?

References:

Chen, F. F. (2008). What happens if we compare chopsticks with forks? The impact of making inappropriate comparisons in cross-cultural research. Journal of Personality and Social Psychology, 95(5), 1005–1018. https://doi.org/10.1037/a0013193

Swami, V., & Barron, D. (2019). Translation and validation of body image instruments: Challenges, good practice guidelines, and reporting recommendations for test adaptation. Body Image, 31, 204-220. https://doi.org/10.1016/j.bodyim.2018.08.014

Fremantle Body Awareness Questionnaire (FreBAQ-general)

- Given that the authors conducted a pilot test, more information on scale development and pilot testing would be extremely beneficial. What did the pilot test involve (e.g., comprehension checks?) How many participants were involved? What revisions were made as a result of this process?

BPI and Brief Pain Catastrophising Scale

- Please provide references for the English and French validations, if both were used. See my previous concern regarding measurement invariance

- Please indicate how raw data were transformed into scale scores (e.g., for BPI pain severity, was a mean of all four items computed?)

- Please provide alpha/McDonald’s omega estimates to indicate levels of internal consistency in the present sample

- Other demographic data collected should be reported under a new subheading.

Results

- Were there any significant differences in key stats between the EFA and CFA samples, or in the distribution of participant characteristics between the samples?

- Please report Bartlett’s test alongside KMO as indicators of data factorability.

- Please report baseline 6-item model fit, in addition to model fit with correlated residuals.

- To assist in assessment of the competing models, I recommend reporting a measure of comparative fit, such as Akaike’s Information Criterion (AIC), and a parsimony-corrected fit index (e.g., the Parsimony Goodness-of-Fit Index [PGFI], Parsimony Normed Fit Index [PNFI]). Likelihood ratio test can also be used to determine whether model adjustments improved model fit.

- Out of interest, did the authors examine the range of body areas that were reported by participants? Were some areas consistently reported, and could the sample be stratified by these areas if desired?

- Could incremental validity be examined via a hierarchical regression, whereby BPI pain severity scores are the criterion variable, and the remaining variables are included in a separate step to the FreBAQ-general, to identify unique variance explained?

Limitations

- The lack of an examination of test-retest reliability is a limitation. Additional analyses (e.g., sensitivity to chance, minimally important clinical difference) would be recommended if the measure is to be used in a clinical capacity, as the authors suggest (e.g., line 341)

Reviewer #2: Thank you very much for the opportunity to review this manuscript (Title: “Exploratory and Confirmatory Factor analysis of the new region-generic version of Fremantle Body Awareness - General Questionnaire”) which addresses the assessment of body awareness in veterans suffering from chronic pain. The authors have validated a self-rating scale that they have developed from a region-specific to a region-generic version. Thus, the Fremantle Body Awareness General Questionnaire is aimed to be used flexibly in different populations regardless of the location of the pain.

The multimodal nature of body awareness necessitates the development of instruments that are specifically tailored for certain diagnostic groups. The statistical analysis is well structured and appropriate methods were used to investigate the instrument’s factorial validity. To the best of our knowledge, the subject of the submitted paper seems not yet to be studied and/or reported in detail elsewhere. However, there are several limitations which lead us to the conclusion that the paper should be revised in its actual form. In the following, major and minor issues of the paper are outlined:

Major issues:

1. In the last paragraph of the introduction, the authors could better explain the advantages of a region-generic version over the region-specific versions of the Fremantle questionnaire by reference to the veteran sample (e.g., pain sites in veterans which are not covered by the available Fremantle scales? Why is a region-generic questionnaire validated in this population? Why reference to social context? etc.).

2. Although not clearly stated by the authors, the text suggests that Cronbach's alpha might be an indicator of a scale's unidimensionality (e.g., lines 126+187). The authors should clarify that alpha is not used to show a scale’s unidimensionality (ref.: Schmitt, N. (1996). Uses and abuses of coefficient alpha. Psychological Assessment, 8(4), 350–353) but to examine internal consistency reliability, which is a prerequisite for validity. Besides, the value of alpha is influenced by the number of items and may therefore not be adequate as an external validation criterion in the present study (reason: reduction of items, e.g., lines 187+269-271). Moreover, Cronbach’s alpha may not be appropriate for use in situations where assumptions of essential tau-equivalence are not met (ref.: Dunn, T. J., Baguley, T., & Brunsden, V. (2014). From alpha to omega: A practical solution to the pervasive problem of internal consistency estimation. British Journal of Psychology, 105(3), 399-412).

3. The authors might consider adding descriptive statistics on the pain localizations expressed by participants (are these localizations not included in the available questionnaire versions?). Such an analysis could further support the need for a generic version.

4. Throughout the manuscript, the authors used the concept of concurrent validity, which is a variant of criterion validity. As the authors noted, concurrent validity is assessed by a positive correlation with an external criterion (usually operationalized as a manifest variable). However, the reviewers believe that the instruments used in this study (BPI, BriefPCS-4) may not be a convincing external criterion for body awareness (surprisingly, this issue was mentioned in the limitations). Instead, the authors might consider referring to convergent validity.

5. Table 2: The authors might consider adding an item analysis (e.g., item-total correlations, item difficulty, item selectivity) and a measure of dispersion (e.g., standard deviation).

6. The authors might consider providing a study flow chart so that there is a better opportunity to retrace participant inclusion/exclusion.

7. Lines 233-242: It appears that the authors used an exploratory approach to confirmatory factor analysis. The pitfalls of confirming the factor structure after modifications in the same sample should be clarified as a limitation. It must be clearly presented that the results of the final model reported in the article are not confirmatory. This would only be the case if the final model were confirmed on a new sample.

Minor issues:

1. The abbreviation “FreBAQ” is simultaneously used for the Fremantle Back Awareness Questionnaire and the Fremantle Body Awareness Questionnaire. This confusing situation should be clarified.

2. The introduction does not provide a clear definition of body awareness, which is also evident in the subsequent discussion of the study findings (line 289-291).

3. Line 99-110: Did participants recruited through Qualtrics receive money or other compensation for participating in the study? If so, what did they receive?

4. Line 122-137: Please clarify if the items were presented in a random or fixed order?

5. Line 199-200: “Consent to participate was provided by 445 Canadian veterans, of which 328 (74%) completed at least 7 of the 9 FreBAQ-body questions on the survey.” -> How many participants completed the 6 item version?

6. Replacing missing values with the mean or simply deleting cases with missing values (lines 162-163) does not comply with the current standard for handling missing values. Authors would be better off using modern imputation techniques when dealing with missing values. Or at least justify why they did so, for example, with very few missing values.

7. There is no clear presentation of which CFA estimation method was used and whether the respective requirements for the methods were met.

8. Analogous to Cronbach’s alpha, factor reliabilities and average variance extracted (AVE) of the factor that can be estimated using CFA should be reported.

9. Lines 234-246: It remains unclear at this point whether both models are inferentially tested against each other. If necessary, this could help to decide in favor of one or the other model.

10. Tables 1+2: The authors should add absolute frequencies, please.

11. Throughout the text, the number of decimal places should be consistent, including in the tables.

The reviewer and co-reviewer declare no conflict of interest.

6. PLOS authors have the option to publish the peer review history of their article (what does this mean?). If published, this will include your full peer review and any attached files.

Reviewer #1: No

Reviewer #2: No

---

## [Author Response · Author response to Decision Letter 0]

7 Dec 2022

Response to Reviewers

Journal Requirements:

RESPONSE: The file has been reformatted to ensure alignment with PLoSOne formatting templates

RESPONSE: We’ve now indicated in the Results section that consent was obtained electronically in accordance with our institution’s ethical guidelines for online survey data collection. More information can be found here https://www.uwo.ca/research/_docs/ethics/hsreb_guidelines/Qualtrics_to_document_informed_consent_28_Sept_2020.pdf

Funded by the Government of Canada and the Chronic Pain Centre of Excellence for Canadian Veterans.

RESPONSE: The Funding statement has been amended accordingly.

RESPONE: These data were collected through a study funded by the Centre of Excellence for Chronic Pain in Canadian military veterans endorsed and facilitated by special interest groups of the Canadian Armed Forces. Ensuring privacy and security of the data was of critical importance to getting buy-in from these groups for the study to run. While not made explicit in the contractual agreement for funding, common sense dictates that non-judicious release of data on items such as proportions of members who occupy non-cis identities, types and rates of traumas experienced, and rates of depression and PTSD are all data that stand to be of particular value to media and advocacy groups while potentially having adverse effects on the CAF. For this reason, out of a strong sense of ethical and moral duty, and acknowledging that sharing of de-identified data sets formed no part of the funding agreement or the participant consent process, we are not releasing these data to a repository for open access. However, we are willing to make the data available for those wishing to confirm our analyses on a request-by-request basis. The main point of contact can be the Western University Office of Research Ethics (ORE) at ethics@uwo.ca

RESPONSE: A caption for the FreBAQ-general attachment has been added to the end of the manuscript

Reviewers' comments:

Reviewer's Responses to Questions

Comments to the Author

Reviewer #1: Thank you for the opportunity to review this work, which reports on the psychometric evaluation of a novel generic version of the Fremantle Body Awareness Questionnaire in a sample of Canadian military veterans with chronic pain. I believe the work will make a useful contribution to literature on body perceptions in chronic pain. However, I have some recommendations for the authors to consider, which mainly centre on the need for greater detail and clarity in reporting. It remains unclear whether the measures were completed in one language (English) or two (English and French), and I have major concerns regarding sematic equivalence and measurement invariance if the method involved the latter.

Introduction

The systematic scoping review by Viceconti et al., 2020 may be useful for guiding some of the introductory comments on body perception and chronic pain, which is currently fairly limited given the content of the questionnaire. There is a large section of the review which focuses on findings from various versions of the Fremantle questionnaire. The review may also be helpful for explaining the later difficulties encountered by the authors with perception of body size for pain-affected body parts.

Reference:

Viceconti, A., Camerone, E. M., Luzzi, D., Pentassuglia, D., Pardini, M., Ristori, D., ... & Testa, M. (2020). Explicit and implicit own's body and space perception in painful musculoskeletal disorders and rheumatic diseases: A systematic scoping review. Frontiers in Human Neuroscience, 14, 83. https://doi.org/10.3389/fnhum.2020.00083

RESPONSE: Thank you for directing us to the Viceconti paper, its inclusion has indeed strengthened the rationale for the study. We’ve included two allusions to the findings of this paper, one in para 3 and one in para 4 of the Introduction. 

Sections: Eligibility criteria and procedure; Participants

- There was some confusion in the organisation of these sections. For example, there was no information in the “eligibility criteria and procedure” section regarding inclusion/exclusion criteria, and some procedural information (e.g., survey contents) was included in the “Participants” section. 

RESPONSE: On re-reading these sections we similarly note the ordering issue. We’ve now re-ordered these two sections so only procedures are listed under the Procedures sub-heading and only participant eligibility criteria are listed under the Participants sub-heading.

I also found it strange that participant N and characteristics were included in the Results section, and not in this dedicated Participants section.

RESPONSE: We respectfully disagree on this item, as the total number of participants recruited and their characteristics are widely considered results of the Methods/Procedures rather than methods themselves. While we’re happy to acknowledge different views on overall manuscript construction, we would have found it strange to have the results of recruitment described in the methods.

- I noted that inclusion criteria included comprehension of English or French. However, it is not clearly stated whether the survey was delivered in both languages. This is a crucial issue because there are best-practice procedures which should be adhered to when preparing a novel translation to ensure that semantic equivalence, and the newly translated questionnaire requires additional psychometric validation (e.g., Swami & Barron, 2019). The English and French versions of the questionnaires may not demonstrate measurement invariance, and it is not appropriate to combine or compare scores from two different versions of the questionnaire until measurement invariance has been established (e.g., see Chen, 2008). This is particularly important for the novel FreBAQ-general, but the point also applies to all other questionnaire measures included within the survey package.

RESPONSE: We are in full alignment with this perspective and acknowledge the value of sound cross-cultural adaptation practices for measurement scales. Sometimes however the realities of what can be accomplished within a funding window require some sacrifices to ‘ideal’ science be made. In this case, while we did have all tools translated by a fluently bilingual translator, we simply did not have the time or funding needed for a full cross-cultural and semantic equivalence study within the 1-year funding window. The reviewer most likely knows and we are only too aware that doing so correctly is a full research study unto itself. This being the Canadian military, offering the survey in both official languages of Canada was a requirement before it would be approved for funding, and we used one of the military’s own translators for that task. By the end of data collection only 9 of the 328 participants who completed at least 7 of the 9 questions used the French version. Our options were to exclude those 9, which would violate the funder’s requirements and is not good practice from an EDI perspective, or include them and acknowledge the potential limit to interpretation. For whatever it’s worth, we did conduct a sensitivity analysis with those 9 responses removed and as I think we’d all expect the results were almost identical, with the communalities off by 1 or 2 100ths of a point. 

However, we now note that we did not in fact dedicate adequate space in the Limitations section to this item, so have now added the following detail: “Finally, the French translation was conducted by a trained bilingual speaker but was not subject to a fulsome cross-cultural translation study to ensure semantic and measurement invariance. While providing a French option was a requirement from an inclusivity perspective in this study of Canadian veterans, we note that only 9 of the 328 respondents in the final sample, or 2.7%, elected to use the French version of the survey. A sensitivity analysis conducted without those 9 responses provided nearly identical results to what has been reported here, though we note that the French version has not undergone formal cross-cultural adaptation and should accordingly be used with caution.”

- What does it mean “Participants completed the balance of the questionnaire independently”? Additional procedural information is required (i.e., did participants complete the survey at home? Was any guidance or interaction with the research team involved?) Furthermore, were participants offered any form of remuneration? How long did the study take to complete on average? Was any debriefing information provided?

References:

Chen, F. F. (2008). What happens if we compare chopsticks with forks? The impact of making inappropriate comparisons in cross-cultural research. Journal of Personality and Social Psychology, 95(5), 1005–1018. https://doi.org/10.1037/a0013193

Swami, V., & Barron, D. (2019). Translation and validation of body image instruments: Challenges, good practice guidelines, and reporting recommendations for test adaptation. Body Image, 31, 204-220. https://doi.org/10.1016/j.bodyim.2018.08.014

RESPONSE: The phrase regarding participants completing the questionnaire independently was intended to indicate that there was no interaction with the research team and that they did complete it on their own wherever they happened to be (at home, work, etc…). From reading the reviewer’s comment it seems that perhaps the word ‘balance’ was misleading as it sounds as though part of the questionnaire was completed under guidance so that word has been removed. Participants were not compensated for their participation. Duration to complete the entire questionnaire is difficult to answer as participants had the opportunity to leave and resume as they pleased, where some would close the window then return while others would just leave it open and come back later. So, the mean time to complete all items on the questionnaire was 62 minutes but this is inclusive of breaks and in no way reflective of how long it would take to complete the FreBAQ items only so we’ve left that aspect out of the manuscript as it is more misleading than helpful.

Fremantle Body Awareness Questionnaire (FreBAQ-general)

- Given that the authors conducted a pilot test, more information on scale development and pilot testing would be extremely beneficial. What did the pilot test involve (e.g., comprehension checks?) How many participants were involved? What revisions were made as a result of this process?

RESPONSE: While the FreBAQ-general was included in the pilot testing of the overall questionnaire/survey, we received no feedback specific to the FreBAQ-general from that process. So, we took that to indicate that those 9 items were adequately understandable to participants and no revisions were made. We describe the pilot testing procedure of the entire survey which is considered best practice for survey-based research. To answer this question more specifically we did not perform a cognitive debriefing component of the FreBAQ-general specifically which may be a nice future study, however we acknowledge that all items were from previously published versions of the FreBAQ and its region-specific alternatives. The original FreBAQ paper of Wand and colleagues did describe a 43-person ‘pilot test’ that while short on descriptive detail one presumes that comprehensibility was a component, and we further note the work of Goossens and colleagues who found a correlation between altered somatosensory processing in the cerebral cortex and FreBAQ responses lending strong evidence of construct validity for the tool which is conceptually the ‘parent’ of the FreBAQ-general described here.

BPI and Brief Pain Catastrophising Scale

- Please provide references for the English and French validations, if both were used. See my previous concern regarding measurement invariance

RESPONSE: We have provided references for BPI and Brief Pain Catastrophising Scale. 

1. (22). Cleeland CS, Ryan KM. Pain assessment: global use of the Brief Pain Inventory. Ann Acad Med Singap. 1994;23: 129–138. 

2. (23). Poundja J, Fikretoglu D, Guay S, Brunet A. Validation of the French version of the brief pain inventory in Canadian veterans suffering from traumatic stress. J Pain Symptom Manage. 2007;33: 720–726. 

3. (25) Walton DM, Mehta S, Seo W, MacDermid JC. Creation and validation of the 4-item BriefPCS-chronic through methodological triangulation. Health Qual Life Outcomes. 2020;18: 124. doi:10.1186/s12955-020-01346-8

- Please indicate how raw data were transformed into scale scores (e.g., for BPI pain severity, was a mean of all four items computed?)

RESPONSE: A the reviewer is likely aware there is little consistency in how BPI pain severity scores tend to be manipulated. We follow the original guidance of Cleeland et al. and use them as a mean of the 4 scores, so that has been added to the manuscript, though alternatively we could have reported them as a simple sum, while others will omit one (often the ‘at its worst’ item) and use only the other 3 for reasons that are not always clear.

- Please provide alpha/McDonald’s omega estimates to indicate levels of internal consistency in the present sample

RESPONSE: While you’ve touched on arguably my least favorite metric of scale function and I dislike including it without additional context, it’s also not a hill I’m willing to die on so have added it to the Methods section for each of the scales used.

- Other demographic data collected should be reported under a new subheading.

RESPONSE: Added

Results

- Were there any significant differences in key stats between the EFA and CFA samples, or in the distribution of participant characteristics between the samples?

RESPONSE: There were no significant differences in terms of sex or length of service. Those in the CFA group were on average 2.4 years older (52.9 vs. 55.3 yrs) than those in the EFA group, which was significant at the p = 0.046 level, arguably by virtue of sample size but of questionable real-world relevance. We’ve added these descriptives to the relevant sections of the Results section.

- Please report Bartlett’s test alongside KMO as indicators of data factorability.

RESPONSE: Added alongside the relevant results

- Please report baseline 6-item model fit, in addition to model fit with correlated residuals.

RESPONSE: Added (line 254-255)

- To assist in assessment of the competing models, I recommend reporting a measure of comparative fit, such as Akaike’s Information Criterion (AIC), and a parsimony-corrected fit index (e.g., the Parsimony Goodness-of-Fit Index [PGFI], Parsimony Normed Fit Index [PNFI]). Likelihood ratio test can also be used to determine whether model adjustments improved model fit.

- Out of interest, did the authors examine the range of body areas that were reported by participants? Were some areas consistently reported, and could the sample be stratified by these areas if desired?

- Could incremental validity be examined via a hierarchical regression, whereby BPI pain severity scores are the criterion variable, and the remaining variables are included in a separate step to the FreBAQ-general, to identify unique variance explained?

RESPONSE: We’ll respond to each of the above 3 comments together. The reviewer is quite right and clearly well-versed in psychometric analyses. There are indeed several additional things we could have done with the data, though they either do not add substantive meaning to the results or interpretation therefore or add extra weight to an already dense paper. Per the comparative fit indicators, those behave as one would expect based on the other indicators we’ve described (e.g., AIC goes from 3273.96 for the uncorrelated 7-item model, to 2755.72 for the 6-item model with residuals of items 1 and 5 correlated). If we included such an indicator, we’d need to also describe it in the Analysis and interpret it in the Discussion, but it does not change the final story so in the interest of parsimony we, respectfully, do not agree with inclusion of those additional metrics unless it becomes a barrier to moving the paper forward. The next two comments are indeed potentially interesting, and again we could well do either. Though again, this is already a dense paper describing a number of analyses. We would like to hold back these additional components for a subsequent manuscript, and for that matter we’ve also got plans to explore the tool through Rasch or IRT for even further detail. Something needs to be held back, and we propose that the analyses described herein are enough to endorse utility of the tool, with the acknowledgment that PROMs always have room for further improvement.

Limitations

- The lack of an examination of test-retest reliability is a limitation. Additional analyses (e.g., sensitivity to chance, minimally important clinical difference) would be recommended if the measure is to be used in a clinical capacity, as the authors suggest (e.g., line 341)

RESPONSE: Similar to our prior response, we agree that if the tool were to be used as a metric of change over time / evaluation then having reliability and responsiveness metrics would be of value. However, this was not a reliability study and we do not have repeat administrations. To acknowledge this comment, we’ve now added a final sentence to the Conclusion section indicating that reliability studies should be a priority for future work.

Reviewer #2: Thank you very much for the opportunity to review this manuscript (Title: “Exploratory and Confirmatory Factor analysis of the new region-generic version of Fremantle Body Awareness - General Questionnaire”) which addresses the assessment of body awareness in veterans suffering from chronic pain. The authors have validated a self-rating scale that they have developed from a region-specific to a region-generic version. Thus, the Fremantle Body Awareness General Questionnaire is aimed to be used flexibly in different populations regardless of the location of the pain.

The multimodal nature of body awareness necessitates the development of instruments that are specifically tailored for certain diagnostic groups. The statistical analysis is well structured and appropriate methods were used to investigate the instrument’s factorial validity. To the best of our knowledge, the subject of the submitted paper seems not yet to be studied and/or reported in detail elsewhere. However, there are several limitations which lead us to the conclusion that the paper should be revised in its actual form. In the following, major and minor issues of the paper are outlined:

Major issues:

1. In the last paragraph of the introduction, the authors could better explain the advantages of a region-generic version over the region-specific versions of the Fremantle questionnaire by reference to the veteran sample (e.g., pain sites in veterans which are not covered by the available Fremantle scales? Why is a region-generic questionnaire validated in this population? Why reference to social context? etc.).

RESPONSE: We’ve now added an allusion to the value of a generic scale in a military context in which many body regions can be affected by injury/trauma (line 93-94). This paragraph has also been extensively revised per the comments from Reviewer 1.

2. Although not clearly stated by the authors, the text suggests that Cronbach's alpha might be an indicator of a scale's unidimensionality (e.g., lines 126+187). The authors should clarify that alpha is not used to show a scale’s unidimensionality (ref.: Schmitt, N. (1996). Uses and abuses of coefficient alpha. Psychological Assessment, 8(4), 350–353) but to examine internal consistency reliability, which is a prerequisite for validity. Besides, the value of alpha is influenced by the number of items and may therefore not be adequate as an external validation criterion in the present study (reason: reduction of items, e.g., lines 187+269-271). Moreover, Cronbach’s alpha may not be appropriate for use in situations where assumptions of essential tau-equivalence are not met (ref.: Dunn, T. J., Baguley, T., & Brunsden, V. (2014). From alpha to omega: A practical solution to the pervasive problem of internal consistency estimation. British Journal of Psychology, 105(3), 399-412).

RESPONSE: We find this perspective very refreshing, as we hold the opinion that Cronbach’s alpha is one of the most over-reported yet misused indices of measurement properties and would prefer not to report it at all under most circumstances. However, Reviewer 1 requested more reporting of alpha, so we’ve done so, but with some degree of trepidation. Nonetheless, if it has to be there then indeed it should be done correctly – we’ve now tested for tau-equivalence and perhaps not surprisingly it was violated for the final (6- and 7-item) versions of the scale. Accordingly, we’ve reported both alpha and omega in Table 4. With a difference of only 2/1000ths of a point neither change the interpretation of scale properties in a meaningful way, but happy to include both in the interest of rigor.

3. The authors might consider adding descriptive statistics on the pain localizations expressed by participants (are these localizations not included in the available questionnaire versions?). Such an analysis could further support the need for a generic version.

RESPONSE: We have added additional supporting information for pain localizations (S1). 

4. Throughout the manuscript, the authors used the concept of concurrent validity, which is a variant of criterion validity. As the authors noted, concurrent validity is assessed by a positive correlation with an external criterion (usually operationalized as a manifest variable). However, the reviewers believe that the instruments used in this study (BPI, BriefPCS-4) may not be a convincing external criterion for body awareness (surprisingly, this issue was mentioned in the limitations). Instead, the authors might consider referring to convergent validity. 

RESPONSE: We agree with the reviewer, and we have revised throughout the manuscript. 

5. Table 2: The authors might consider adding an item analysis (e.g., item-total correlations, item difficulty, item selectivity) and a measure of dispersion (e.g., standard deviation).

RESPONSE: Again, this is already a dense paper describing a number of estimates and we do not want to overburden readers. 

6. The authors might consider providing a study flow chart so that there is a better opportunity to retrace participant inclusion/exclusion.

RESPONSE: Lines 203-204 “Electronic online consent to participate was provided by 445 Canadian veterans, of which 328 (74%) completed at least 7 of the 9 FreBAQ-body questions on the survey”

7. Lines 233-242: It appears that the authors used an exploratory approach to confirmatory factor analysis. The pitfalls of confirming the factor structure after modifications in the same sample should be clarified as a limitation. It must be clearly presented that the results of the final model reported in the article are not confirmatory. This would only be the case if the final model were confirmed on a new sample.

Minor issues:

1. The abbreviation “FreBAQ” is simultaneously used for the Fremantle Back Awareness Questionnaire and the Fremantle Body Awareness Questionnaire. This confusing situation should be clarified.

RESPONSE: Clarified

2. The introduction does not provide a clear definition of body awareness, which is also evident in the subsequent discussion of the study findings (line 289-291).

RESPONSE: Introduction has been revised based on reviewer’s 1 suggestions

3. Line 99-110: Did participants recruited through Qualtrics receive money or other compensation for participating in the study? If so, what did they receive?

RESPONSE: No financial or other incentives were offered to the study participants.

4. Line 122-137: Please clarify if the items were presented in a random or fixed order?

RESPONSE: Fixed. Lines 129-130.

5. Line 199-200: “Consent to participate was provided by 445 Canadian veterans, of which 328 (74%) completed at least 7 of the 9 FreBAQ-body questions on the survey.” -> How many participants completed the 6 item version?

RESPONSE: 74% (n=328)

6. Replacing missing values with the mean or simply deleting cases with missing values (lines 162-163) does not comply with the current standard for handling missing values. Authors would be better off using modern imputation techniques when dealing with missing values. Or at least justify why they did so, for example, with very few missing values.

RESPONSE: There is no clear consensus which handling missing values technique is the most appropriate. To handle missing we selected to use a simple mean imputation technique since it was unclear if the missing values were violating the MAR assumption or not (Donders 2006, Clin Epi Journal). 

7. There is no clear presentation of which CFA estimation method was used and whether the respective requirements for the methods were met. 

RESPONSE: Lines 181-187 “Standard goodness-of-fit indicators and fit residuals were used to interpret these results: comparative fit index (CFI) > 0.90 [33], Tucker-Lewis Index (TLI) > 0.90, root mean square error of approximation (RMSEA) < 0.08 [34], and standardized root mean square residual (RSMR) < 0.05 [34]. Where the indicators suggested sub-optimal fit, residuals and modification indices were explored to identify potentially problematic items or location dependence between two or more items. Where a potentially problematic item was identified, it was removed and the model retested, followed by re-evaluation of the remaining items in the EFA with the first random set.”

8. Analogous to Cronbach’s alpha, factor reliabilities and average variance extracted (AVE) of the factor that can be estimated using CFA should be reported.

RESPONSE: Please see our response above for the reporting of additional estimates.

9. Lines 234-246: It remains unclear at this point whether both models are inferentially tested against each other. If necessary, this could help to decide in favor of one or the other model.

RESPONSE: The models were not tested against each other. 

10. Tables 1+2: The authors should add absolute frequencies, please.

RESPONSE: added.

11. Throughout the text, the number of decimal places should be consistent, including in the tables.

RESPONSE: addressed.

The reviewer and co-reviewer declare no conflict of interest.

6. PLOS authors have the option to publish the peer review history of their article (what does this mean?). If published, this will include your full peer review and any attached files.

Do you want your identity to be public for this peer review? For information about this choice, including consent withdrawal, please see our Privacy Policy.

Reviewer #1: No

Reviewer #2: No

---

## [Decision Letter · Decision Letter 1]

22 Dec 2022

PONE-D-22-16116R1Exploratory and Confirmatory Factor analysis of the new region-generic version of Fremantle Body Awareness - General QuestionnairePLOS ONE

Dear Dr. Walton,

Thank you for submitting your manuscript to PLOS ONE. One reviewer is now happy to recommend publication but the other reviewer has reiterated their opinion that reporting competing model fits (e.g. AIC), body areas, and incremental validity would be valuable for the publication, and I agree with this.

We look forward to receiving your revised manuscript.

Kind regards,

Jane Elizabeth Aspell, PhD

Academic Editor

PLOS ONE

Journal Requirements:

Reviewers' comments:

Reviewer's Responses to Questions

**Comments to the Author**

1. If the authors have adequately addressed your comments raised in a previous round of review and you feel that this manuscript is now acceptable for publication, you may indicate that here to bypass the “Comments to the Author” section, enter your conflict of interest statement in the “Confidential to Editor” section, and submit your "Accept" recommendation.

Reviewer #1: (No Response)

Reviewer #2: All comments have been addressed

2. Is the manuscript technically sound, and do the data support the conclusions?

Reviewer #1: Yes

Reviewer #2: Yes

3. Has the statistical analysis been performed appropriately and rigorously? 

Reviewer #1: Yes

Reviewer #2: Yes

4. Have the authors made all data underlying the findings in their manuscript fully available?

Reviewer #1: (No Response)

Reviewer #2: No

5. Is the manuscript presented in an intelligible fashion and written in standard English?

Reviewer #1: (No Response)

Reviewer #2: Yes

6. Review Comments to the Author

Reviewer #1: Thank you for the opportunity to review this work again. The authors have responded to all of my comments thoroughly, and I have nothing new to add at this stage.

On my points on competing model fits (reporting AIC etc.), body areas, and incremental validity, I respectfully disagree with the authors that including these analyses would be burdensome for readers. Plos has no word count limits, and I feel these analyses would make a useful contribution to the present paper. If the authors are hoping to publish further works, best practice would suggest collecting new data to further validate the measure in a new sample (particularly as there were some exploratory modifications in the CFA analyses), rather than publishing more analyses on the same dataset (i.e., dual publication issues). However, I leave this issue to the editor's discretion.

Reviewer #2: We thank the authors for adressing the issues. The manuscript significantly improved and can be published in the present form.

7. PLOS authors have the option to publish the peer review history of their article (what does this mean?). If published, this will include your full peer review and any attached files.

Reviewer #1: No

Reviewer #2: No

---

## [Author Response · Author response to Decision Letter 1]

24 Jan 2023

We would like to thank the reviewers for the valuable feedback and comments.

Response: As reviewer #1 pointed out there are indeed several additional things we could have done with the data, though they either do not add substantive meaning to the results or interpretation therefore or add extra weight to an already dense paper. Per the comparative fit indicators, those behave as one would expect based on the other indicators we’ve described (e.g., AIC goes from 3273.96 for the uncorrelated 7-item model, to 2755.72 for the 6-item model with residuals of items 1 and 5 correlated). If we included such an indicator, we’d need to also describe it in the Analysis and interpret it in the Discussion, but it does not change the final story so in the interest of parsimony we, respectfully, do not agree with inclusion of those additional metrics unless it becomes a barrier to moving the paper forward. Reviewer also pointed out that it is to the Editor’s discretion. Though again, this is already a dense paper describing a number of analyses. We would like to hold back these additional components for a subsequent manuscript, and for that matter we’ve also got plans to explore the tool through Rasch or IRT for even further detail. Something needs to be held back, and we propose that the analyses described herein are enough to endorse utility of the tool, with the acknowledgment that PROMs always have room for further improvement.

---

## [Decision Letter · Decision Letter 2]

28 Feb 2023

Exploratory and Confirmatory Factor analysis of the new region-generic version of Fremantle Body Awareness - General Questionnaire

PONE-D-22-16116R2

Dear Dr. Walton,

We’re pleased to inform you that your manuscript has been judged scientifically suitable for publication and will be formally accepted for publication once it meets all outstanding technical requirements.

Kind regards,

Jane Elizabeth Aspell, PhD

Academic Editor

PLOS ONE

Additional Editor Comments (optional):

Reviewers' comments:

Reviewer's Responses to Questions

**Comments to the Author**

1. If the authors have adequately addressed your comments raised in a previous round of review and you feel that this manuscript is now acceptable for publication, you may indicate that here to bypass the “Comments to the Author” section, enter your conflict of interest statement in the “Confidential to Editor” section, and submit your "Accept" recommendation.

Reviewer #1: All comments have been addressed

2. Is the manuscript technically sound, and do the data support the conclusions?

Reviewer #1: Yes

3. Has the statistical analysis been performed appropriately and rigorously? 

Reviewer #1: Yes

4. Have the authors made all data underlying the findings in their manuscript fully available?

Reviewer #1: Yes

5. Is the manuscript presented in an intelligible fashion and written in standard English?

Reviewer #1: Yes

6. Review Comments to the Author

Reviewer #1: I continue to respectfully disagree with the authors regarding the withholding of additional analyses for future papers; I think they would strengthen the present paper, and would not overburden the reader. However, I acknowledge that the statistics I requested are not essential for the publication of this work, and I will recommend publication on this basis to avoid further delays.

7. PLOS authors have the option to publish the peer review history of their article (what does this mean?). If published, this will include your full peer review and any attached files.

Reviewer #1: No

---

## [Editor Report · Acceptance letter]

14 Mar 2023

PONE-D-22-16116R2 

Exploratory and Confirmatory Factor analysis of the new region-generic version of Fremantle Body Awareness - General Questionnaire 

Dear Dr. Walton:

I'm pleased to inform you that your manuscript has been deemed suitable for publication in PLOS ONE. Congratulations! Your manuscript is now with our production department. 

Kind regards, 

on behalf of

Dr. Jane Elizabeth Aspell 

Academic Editor

PLOS ONE